# Enhancing the Biodiesel Production by Improving the Yield of Lipids in Wild Strain by Inducing Nitrogen Ion Mutation in *Rhodotorula mucilaginosa*

**Joseph Antony Sundarsingh Tensingh [1,2] and Vijayalakshmi Shankar [1,2,]**

1    School of Biosciences and Technology, Vellore Institute of Technology, Vellore 632014, India
2    CO$_2$ Research & Green Technologies Center, Vellore Institute of Technology, Vellore 632014, India
*    Correspondence: vijimicro21@gmail.com

**Abstract:** The overconsumption of energy results in the depletion of fossil fuels. Generally, biodiesels are produced from wastes of animal fats and vegetable oils. In this study, we have tried to produce biodiesel from both the wild strain and ion beam mutated strain and compared the concentration of lipids produced from both the strains and their properties. Lipids were extracted from microbes using the Bligh and Dyer method and analyzed using gas chromatography and mass spectrophotometry (GCMS) and Fourier-transform infrared (FTIR) spectroscopy. Extracted lipids (free fatty acids) were converted into biodiesel (fatty acid methyl esters) using a base catalyst. The end product biodiesel was characterized and analyzed based on ASTM standards.

**Keywords:** microorganism; ion beam radiation; GC-MS; FTIR; biodiesel

## 1. Introduction

Oleaginous microorganisms can produce and accumulate lipids from their cell biomass weight of around 20%. The lipids that are obtained from microorganisms are named single-cell oils [1]. The production of lipids in oleaginous microorganisms are mainly based on their gene; thus, it differs according to various species and even between the same species of different strains. The major components of lipids are triacylglycerols (TAGs) and stearyl esters (SE). The three major components of triacylglycerol are made up of fatty acid chains in which glycerol acts as the main component, and that can be utilized in biodiesel production [2]. The major example of triacylglycerol-based biofuel is biodiesel that requires very few volumes of feedstock, the biodiesel production cost is low compared to fossil fuel production. In this research, we observed that oleaginous microorganisms such as algae, fungi, and yeasts are the most microorganisms utilized in lipid biotechnology research [3]. Microalgae have the capacity to produce high potential lipids with approximately 70% yield from their dry cell biomass weight according to lipid-producing oleaginous microorganisms [4]. However, the yield of lipids produced from microalgae differs from microalgae and strains of the same species; therefore, it is mainly dependent on their culture conditions [5]. Likewise, unicellular microorganisms such as oleaginous yeasts have the capacity to produce higher concentrations of lipids. Some of the species of oleaginous yeast are *Cryptococcus*, *Yarrowia*, *Rhodotorula*, *Candida*, *Lipomyces*, *Rhodosporidium*, and *Trichosporon*, and they are capable of producing lipids in their biomass under stress environments while culturing in a wide range of carbon source, which yields up to 80% lipids [6].

Mutations in microbes can occur by two major methods such as random mutagenesis and metabolic engineering [7]. Different product yields and their accumulation cannot be majorly modified by metabolic engineering due to their specific genetic sequence and metabolic cycle in the microbe. Hence, random mutagenesis plays an important role in causing mutation in microorganisms [8]. There are many artificial mutagenesis methods

that target developing the rate of mutation using physical or chemical mutagens. Ionizing radiation is a type of physical mutation that includes helium, ion, and heavily charged particles, which are mainly used as a mutagen for mutating microbes [9].

Random mutagenesis in yeast is mainly done by using physical mutagens that cause modifications in the DNA sequence by addition or deletion of ATGC components in DNA [10]. Physical mutagens are processed by exposing microbial cell culture to radiation such as ion beam, UV, nuclear, X-rays, and γ-rays. Heavier charged particles are used in ion beam radiation, which causes DNA damage and produces high mutation rates when compared with X-rays and γ-rays [11]. Another method of ion beam radiation is passing low-energy nitrogen ions, which cause mutation on the surface of microbial cells. Afterwards, the ion radiation produces free radicals inside the cell that causes mutation in DNA [12]. Accumulation of lipids in the cells is identified using staining techniques. The cells are stained using lipophilic dyes, such as Nile red staining [13].

The molecular pathway of synthesizing triacylglycerol in microalgae and yeasts starts with acetyl-CoA formation. The produced acetyl-CoA from the pyruvate was catalyzed using pyruvate dehydrogenase complex, and then acetyl-CoA is carbonylated using acetyl-CoA carboxylase to produce malonyl-CoA [14]. Then, the next process of producing lipids is transferring malonyl-CoA to the Acyl carrier protein, which is catalyzed using the fatty acid synthase complex [15]. We tried to mutate the *R. mucilaginosa* strain by using ion-beam irradiation and studied it for lipid enhancement for the production of biodiesel.

This work mainly focuses on the production of biodiesel from an ion beam radiation mutated microbe to improve the quality and quantity of biodiesel according to ASTM standards.

## 2. Materials and Methods

### 2.1. Isolation of Microbes

The soil was collected from the VVD coconut oil factory in Thoothukudi, Tamil Nadu. The collected soil proceeded to the serial dilution method aseptically using deionized water inside a laminar airflow chamber [16]. The sample was diluted in different concentrations at the ratio of $10^{-9}$, $10^{-8}$, $10^{-7}$, $10^{-6}$, $10^{-5}$, $10^{-4}$, $10^{-3}$, $10^{-2}$, and $10^{-1}$. Then, the diluted samples were cultured in yeast peptone dextrose broth media at the ratio of 1:2:2; therefore, yeast extract (10 g), peptone (20 g), and dextrose (20 g) were mixed in 1 L sterile double distilled water and in nutrient Agar media [beef extract (3.0 gm), peptone (5 g), NaCl (0.5 g), and agar (15 g) mixed with 1 L double distilled water] [17]. Approximately ten fungal and bacterial colonies were isolated from $10^{-6}$ and $10^{-7}$ dilutions from the soil sample. Among the isolated microbes, a few microorganisms found to possess a high growth rate with maximum lipid-yielding potential was selected for further analysis. These selected microbes were stained using Nile red staining and screened for their lipid concentration [18].

### 2.2. Ion Beam Radiation

The ion beam radiative instrument consists of eight segments as represented in Figure 1. The functioning principle of the instrument was followed by various steps: the tungsten filament present in the ion source emits charged particles such as free electrons, and they get accelerated; then, the ion beam produced from the ion source chamber is extracted by an extraction electrode and screened by mass analyzer [19]. The extraction electrode accelerates ions up to 50 keV. Then the ions pass through the acceleration tube and increase energy. Afterwards, the ion beam is ready for sample implantation. The vacuum chamber plays an important role by focusing the ion beam on the target sample and controlling of scattering and collision of beams [20].

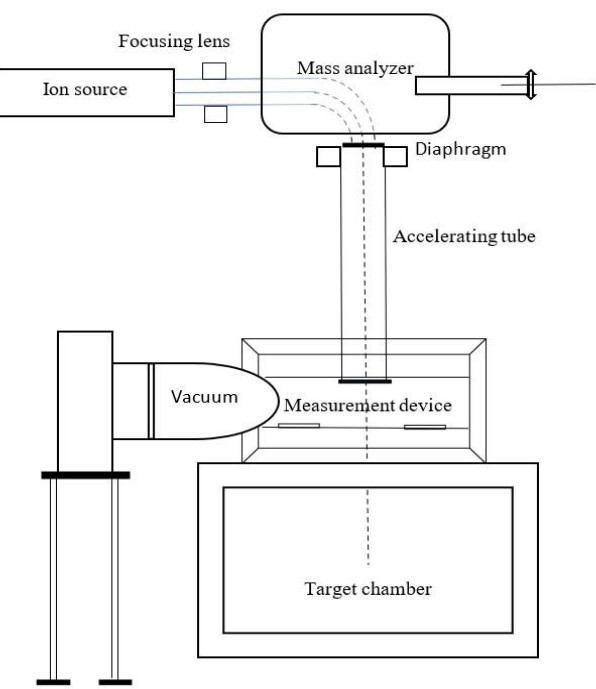

**Figure 1.** Schematic representation of ion beam radiation.

### 2.3. Physical Mutation

The highest lipid-yielding strain from the isolated microorganism has been mutated using nitrogen ion beam radiation by 10 keV (kilo electron volts). About 1 mL of isolated microbial culture was taken from the log phase and filtered using sterile filtrate; then, the sample was mixed with 15% of glycerin, which acts as a guard to cells [21]. About 0.1 mL of the processed sample solution was suspended on a sterile plate and placed in a hot air oven for drying. Next, the sample was radiated using nitrogen ions [22]. The radiation of the ion beam was intended to mutate the sample with different nitrogen concentrations, $0$ $N^+/cm^2$, $0.5 \times 10^{14}$ $N^+/cm^2$, $1.0 \times 10^{14}$ $N^+/cm^2$, $1.5 \times 10^{14}$ $N^+/cm^2$, $2.0 \times 10^{14}$ $N^+/cm^2$, $2.5 \times 10^{14}$ $N^+/cm^2$, and $3.0 \times 10^{14}$ $N^+/cm^2$. A control sample was placed in a vacuum chamber without radiation. After mutating the samples with different nitrogen concentrations, the mutated samples were rinsed using a sterile saline solution containing 8.5 g of NaCl in 1 L of double distilled water [23].

### 2.4. Selection of Nitrogen Ion Mutated Strain

The nitrogen mutated strain was cultured on yeast extract peptone dextrose agar medium for 72 h at 29 °C to analyze the survivability of mutated strains in different nitrogen ion radiation concentrations [24]. Then, the produced colony was inoculated in YEPD broth, placed inside the rotary shaker, and maintained at 29 °C and 180 rotations per minute for 24 h. Next, 1 mL of broth culture was stained with Nile red staining and analyzed for maximum lipid production. The maximal lipid-producing mutated strain was named MI-1.

### 2.5. Determination of Mutation Rate

The mutated strain was identified by the colonies produced after the radiation of the ion beam to the sample. The colonies were cultured, and the lipids extracted from the strains were compared with the wild strain (Strain without mutation) [25]. The criterion for mutants was defined as the strains whose oil production was 10% higher/lower than that

of the control strain and were considered positive/negative mutants. The mutation rate was determined by the following formula:

$$\text{Mutation rate} = \frac{M1 - M0}{N}$$

where $M1$ indicates the colonies produced in the mutated strain by ion beam radiation, $M0$ denotes the colonies cultured in wild strain plates, and $N$ denotes total colonies cultured in both wild and mutated strain [26]. Therefore, the mutation rate was analyzed by averaging the colonies produced by both wild and mutated strains. Mutant selection colonies larger than unirradiated colonies were picked for re-screening. After being cultured in seed medium under the condition of 30 °C and 170 rpm for 24 h, the samples were inoculated into 20 mL N-limited fermentation medium at 5% and cultured for 7 days under the same conditions.

### 2.6. Molecular Sequence Analysis of Selected Strains

A molecular identification method was performed to classify the species of the isolated microbe. The isolation of the genomic DNA of the microorganism was performed using the Qiagen DNeasy Kit. A large ribosomal subunit was amplified using the standard PCR reaction method [27]. The primers for the wild strain and ion beam mutated strain were performed using the LSU region, and when using the LR7 and 5.8 SR primers, the fragments are amplified. After amplification, using a gene O-spin purification kit, the fragments of DNA are purified and then directly sequenced and processed to GenBank.

### 2.7. Culture and Growth Conditions of Wild and Nitrogen Ion Mutant Strain

The wild and nitrogen ion mutated strain was inoculated in two different 500 mL conical flask containing 250 mL of yeast extract peptone dextrose broth culture. The culture was standardized and placed on a rotary shaker and maintained at 30 °C and 170 rpm for one day. Then, the culture was taken from broth media and transferred into a 2 mL Eppendorf tube and centrifuged for 6 min at 13,000 rpm, and the pellet was taken aseptically [28]. Next, the pellet was weighed to analyze the biomass weight cultured in broth every 24 h.

### 2.8. Determination of Lipid and Biomass Yield

Approximately 10 mL of broth culture was aspirated aseptically and transferred to a 15 mL centrifuge tube. Then, the culture is centrifuged for 10 min at 4 °C and 3000 rpm. Next, the biomass was collected from the centrifuge tube, washed using sterilized double distilled water, and air dried using a hot air oven for 2 h at 80 °C [29]. The yield of the biomass was expressed in g/L [24]. The lipid was extracted using the Bligh and Dyer method with the biomass produced from the wild strain VS3 and nitrogen ion mutated strain MI-1. At every 24 h interval, the biomass yield and lipid content were recorded. To check the lipid-producing ability of the mutant microbe after subculturing, the mutant strain was subcultured about 30 times, and the lipid concentration was analyzed at every subculturing. The percentage of lipid produced from biomass is calculated [30] as follows:

$$\text{Lipid percentage} = \frac{\text{Initial biomass weight} - \text{Final biomass weight}}{\text{Total weight of the sample}} \times 100$$

### 2.9. Extraction of Lipids from the Wild and Nitrogen Ion Mutant Strain

Lipid was extracted using the Bligh and Dyer method. This method of extracting lipids was used widely, and it is also a common method for the extraction of total lipids [30]. In this process, the biomass of the wild and nitrogen ion beam mutated strains was centrifuged, and the pellet was dried and mixed with a 2:1:1 (*v/v/v*) ratio of methanol:chloroform:water and mixed vigorously for 10 min [31]. Then, 0.85% of potassium chloride was mixed with the solution and made up to 2:2:2 (*v/v/v*). Next, the mixture was again blended using

a vortex machine for 10 min [32]. Then, the solvent was kept on the stand without any disturbance, and the cell debris was discarded. Finally, the lipid was removed slowly and assessed for total lipid content [33].

### 2.10. GCMS Analysis of Extracted Lipids

Approximately 200 µL of extracted lipids from both wild and nitrogen ion irradiated strains was transferred into a conical flask containing 4 mL of 0.5 M KOH solution and mixed gently by shaking. The flask attached to the reflux condenser containing stock solution was gently placed in a boiling water bath for 15 min by periodical shaking [34]. Then, 1.6 mL of methanolic HCL acid was added to the flask and again boiled for 25 min. Next, the reflux condenser was gently removed from the flask, and the solvent was placed at room temperature to cool. A total of 8 mL of deionized distilled water was added, and the solvent was mixed gently by shaking. Then, the solution was mixed continuosly for 2 min by adding 6 mL of n-Hexane to extract the fatty esters [35]. The process was repeated three times to extract the whole FAME from the sample for analysis in gas chromatography and mass spectroscopy. To analyze the lipid content present in the extracted FAME from both wild and nitrogen ion mutated strains, an Agilent 6890 gas chromatograph instrument was used. It contains an EC-5 column of 250 µ I.D., 0.25 µ film thickness, and a 2 mm direct injector liner. The split injection attached to the instrument was used for sample loading at the ratio of 10:1. It starts automatically when the temperature reaches 35 °C in the oven and is kept for 2 min. Afterwards, it rises at 20 °C/min up to 300 °C for 5 min. Therefore, the helium carrier gas flow rate was constantly programmed at 2 mL per min. A benchtop mass spectrometer JEOL GCmate II with a double focusing magnetic sector operating electron ionization (EI) mode with the software TSS-2000 was used for analyses [36].

### 2.11. FTIR Analysis of Extracted Lipids

In this analytical method, 460 plus FTIR spectrometers (JASCO, Easton, MA, USA) in the range of $cm^{-1}$ are used to characterize the lipid quality by analyzing the functional groups [37]. Extracted lipids were individually combined with KBr, formed a pellet, and placed on an FTIR plate. Then, the composite was analyzed in a spectrum ranging from 400 to 4000.

### 2.12. Transesterification

The extracted lipids from both wild and nitrogen ion mutant strains were taken aseptically and transferred into a sterilized conical flask. Preparation of the catalyst was done by adding 4 g of NaOH pellet in 100 mL methanol. Approximately 10 mL of freshly prepared catalyst was added to 1 mL of extracted lipids from the wild and nitrogen ion irradiated strain ratio of 1:10 (lipid:catalyst) in two different conical flasks. The mixture was then heated at 60 to 70 °C for 90 min in a thermo-magnetic stirrer plate. Then, the solution was poured into a separating funnel and kept for one day on a clean surface. The produced biodiesel was collected from the separating funnel, and the properties of biodiesel were analyzed [38].

### 2.13. Properties of Biodiesel

The properties of biodiesel were analyzed to check the efficiency and performance of the synthesized biodiesel. The important properties of biodiesel to test are density, kinematic viscosity, specific gravity, fire point, flash point, calorific value, pour point, cetane number, and acid value [39]. All these mentioned parameters were tested in the produced biodiesel from both wild and nitrogen ion mutant strains. The concordant value was noted and referred to ASTM standards.

## 3. Results

### 3.1. Screening of Maximum Lipid Production

Approximatly 10 bacterial and fungal colonies were isolated from the soil sample and screened for high lipid yield. Out of these 10 colonies, one microbial strain, SB2, exhibited maximum lipid production ability. Table 1 represents the concentration of lipids produced from bacterial strains (SB) and fungal strains (SF). The particular SB2 was chosen for further study.

**Table 1.** Concentration of lipids extracted from isolated microbes.

| Strain | Lipid Concentration (g/L) | Strain | Lipid Concentration (g/L) |
| --- | --- | --- | --- |
| SB1 | $0.27 \pm 0.22$ | SF1 | $0.26 \pm 0.37$ |
| SB2 | $0.58 \pm 0.14$ | SF2 | $0.35 \pm 0.15$ |
| SB3 | $0.05 \pm 0.26$ | SF3 | $0.12 \pm 0.42$ |
| SB4 | $0.38 \pm 0.32$ | SF4 | $0.15 \pm 0.39$ |
| SB5 | $0.26 \pm 0.19$ | SF5 | $0.13 \pm 0.26$ |
| SB6 | $0.15 \pm 0.16$ | SF6 | $0.23 \pm 0.32$ |
| SB7 | $0.08 \pm 0.21$ | SF7 | $0.14 \pm 0.41$ |
| SB8 | $0.20 \pm 0.5$ | SF8 | $0.14 \pm 0.24$ |
| SB9 | $0.13 \pm 0.35$ | SF9 | $0.3 \pm 0.31$ |
| SB10 | $0.14 \pm 0.17$ | SF10 | $0.13 \pm 0.27$ |

### 3.2. Survival Rate of Nitrogen Mutated Strain

The survival rate of the wild strain exposed to nitrogen ion radiation of different concentrations showed a sloppy bend curve with an increasing dosage of ions. When the strain was mutated from 0 ions/cm$^2$ to $0.5 \times 10^{14}$ ions/cm$^2$, there was a 20 percent decrease in the survival rate. When the dosage of ion radiation increased from $0.5 \times 10^{14}$ ions/cm$^2$ to $1.0 \times 10^{14}$ ions/cm$^2$, a 40 percent decrease in the survival rate was identified. Additionally, increasing the dosage of ions reduced the survival rate of the wild strain. The survival rate of the microbe when irradiated with nitrogen ion is illustrated in Figure 2. The mutation performed on the wild strain was implanted by nitrogen ions, the survivability of the mutant strain was stable at $1.5 \times 10^{14}$ ions/cm$^2$ to $2.0 \times 10^{14}$ ions/cm$^2$, and the highest positive mutation rates were screened and analyzed. Thus, the colonies that survived at $1.5 \times 10^{14}$ ions/cm$^2$ were selected and cultured. Similarly, Li Shichang et al. (2013) also mutated *R. glutins* using the low ion implantation method and showed a maximum mutation at $2.6 \times 10^{13}$ ions/cm$^2$, which was almost similar to our study [30]. Yuanyuan Cao et al. (2010) also mutated *Geotrichum robustum* using nitrogen ion radiation of different concentrations and received the same mutation rate in the range of $2.0 \times 10^{14}$ ions/cm$^2$ in this study [29]. The mutation rate of the wild strain exposed to N+ at the energy of 10 keV, the correlation between the mutation rate, and the dose-dependent survival rate has been illustrated in Figure 3. At lower doses ($0.5 \times 10^{14}$ ions/cm$^2$), the negative and positive mutants were found to be minimal. Maximal positive mutations were identified at $2.0 \times 10^{14}$ ions/cm$^2$, and a further increase in the nitrogen ions dose resulted in less mutation. All the experiments were done in triplicates.

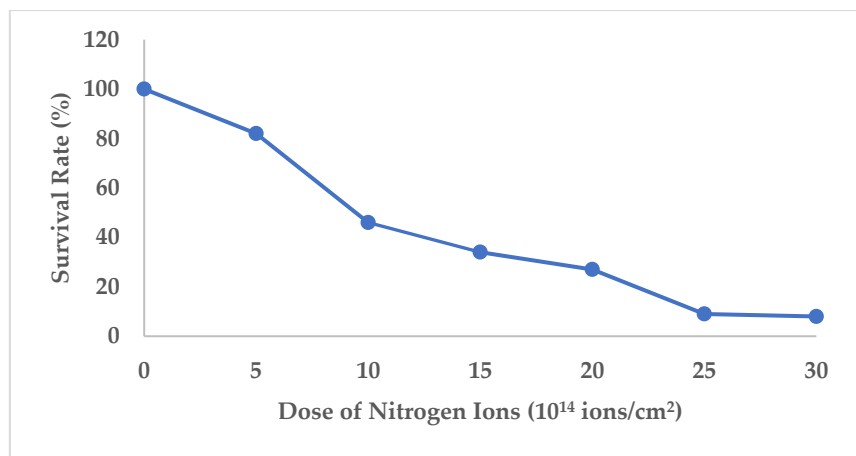

**Figure 2.** A graphical representation of the wild strain *Rhodotorula mucilaginosa* VS3 survivability on different nitrogen ion beam radiation concentrations.

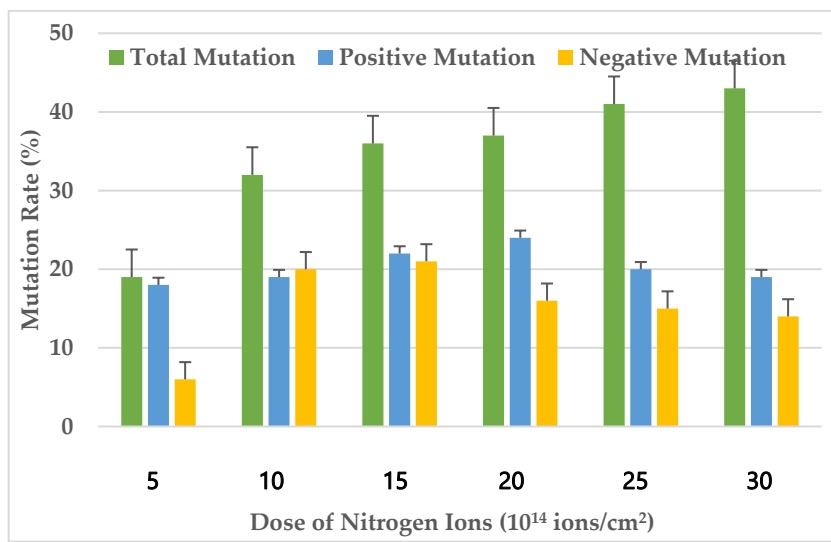

**Figure 3.** The mutation rate of the wild strain exposed to N+ at the energy of 10 keV. Data were average values of three independent experiments. Error bars indicate the SE of mean values.

### 3.3. Molecular Identification of the Isolated Microorganism

The isolation of DNA from the selected microbe was amplified using PCR. The amplified sequence was submitted to GenBank, the accession number received was OL635991, and the microorganism was identified as *Rhodotorula mucilaginosa* VS3. The same procedure was followed for the ion beam mutated strain, and the sequence was submitted to GenBank to understand the mutations that occurred in the DNA of the specific microorganism. The ion beam mutated strain was named *Rhodotorula mucilaginosa* MI-1, and the accession number was OL658826. The phylogenetic tree of *Rhodotorula mucilaginosa* was retrieved from BLAST as represented in Figure 4.

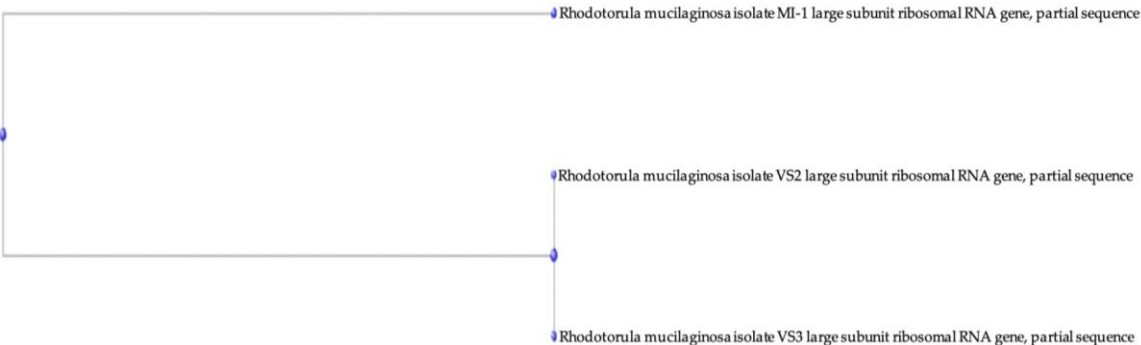

**Figure 4.** Phylogenetic tree analysis of the wild and nitrogen ion mutated strain.

### 3.4. Lipid and Biomass from the VS3 and MI-1 Strain

Biomass production was determined gravimetrically by centrifuging the broth culture of *Rhodotorula mucilaginosa* VS3 and MI-1 and oven-drying the pellet. By using a vacuum weighing machine, the pellet was weighed. Lipids from both VS3 and the nitrogen ion irradiated strain MI-1 have been extracted using the Bligh and Dyer method. Production of the VS3 strain and MI-1 biomass is represented in Table 2 and Figure 5. Additionally, the lipid concentration from both the wild and nitrogen ion mutated strain MI-1 is illustrated in Table 3 and Figure 6. The lipid production ability of the mutant strain after several subcultures was tested in up to 30 subcultures. We found that even after 30 subcultures, the mutant strain has the ability to withstand about 90% of the lipid-production capability (Table 4).

**Table 2.** A tabular representation of the production of biomass.

| S.No | Time (h) | Biomass of Wild Strain *R. mucilaginosa* VS3 (g/L) | Biomass of Nitrogen Ion Mutated Strain *R. mucilaginosa* MI-1 (g/L) |
|---|---|---|---|
| 1 | 0 | 0 | 0 |
| 2 | 12 | $2.7 \pm 0.3$ | $6.2 \pm 0.7$ |
| 3 | 24 | $3.8 \pm 1.1$ | $9.3 \pm 0.1$ |
| 4 | 36 | $5.4 \pm 0.9$ | $12.2 \pm 1.9$ |
| 5 | 48 | $7.5 \pm 0.6$ | $18.5 \pm 2.1$ |
| 6 | 60 | $9.6 \pm 1.8$ | $23.2 \pm 1.4$ |
| 7 | 72 | $11.8 \pm 1.9$ | $26.5 \pm 0.8$ |
| 8 | 84 | $13.5 \pm 0.5$ | $32.3 \pm 0.4$ |
| 9 | 96 | $15.8 \pm 1.2$ | $36.2 \pm 1.5$ |
| 10 | 108 | $17.5 \pm 0.8$ | $39.7 \pm 1.7$ |
| 11 | 120 | $18.7 \pm 1.6$ | $40.2 \pm 0.4$ |

**Table 3.** A tabular representation of the extraction of lipids.

| S.No | Time (h) | Lipid Extracted from Strain *R. mucilaginosa* VS3 (%) | Lipid Extracted from Nitrogen Ion Mutated Strain *R. mucilaginosa* MI-1 (%) |
|---|---|---|---|
| 1 | 0 | 0 | 0 |
| 2 | 12 | $3.5 \pm 0.5$ | $4.5 \pm 1.4$ |
| 3 | 24 | $9.5 \pm 1.4$ | $14.6 \pm 1.9$ |
| 4 | 36 | $12.4 \pm 0.26$ | $17.5 \pm 0.7$ |
| 5 | 48 | $17.4 \pm 1.8$ | $23.1 \pm 0.3$ |
| 6 | 60 | $21.6 \pm 0.7$ | $32.6 \pm 1.1$ |
| 7 | 72 | $28.5 \pm 1.1$ | $37.8 \pm 0.9$ |
| 8 | 84 | $32.5 \pm 0.9$ | $42.3 \pm 1.3$ |
| 9 | 96 | $37.3 \pm 1.6$ | $45.6 \pm 0.5$ |
| 10 | 108 | $40.9 \pm 0.7$ | $48.9 \pm 1.7$ |
| 11 | 120 | $41.3 \pm 0.8$ | $51.2 \pm 0.8$ |

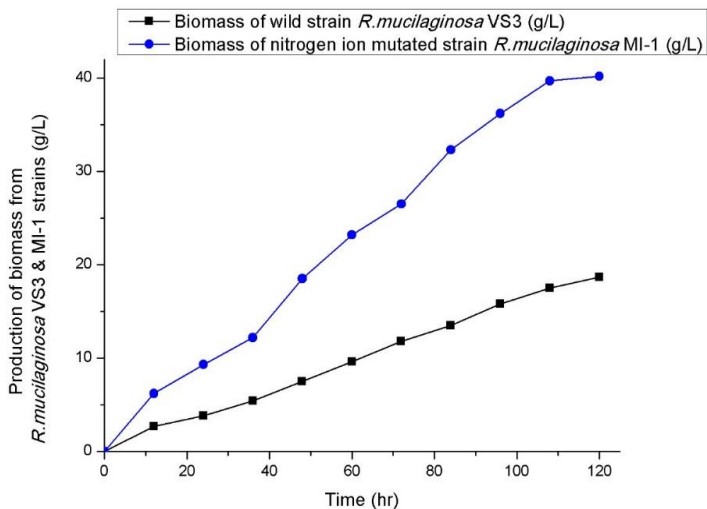

**Figure 5.** Production of biomass from the VS3 and MI-1 strains of *R. mucilaginosa*.

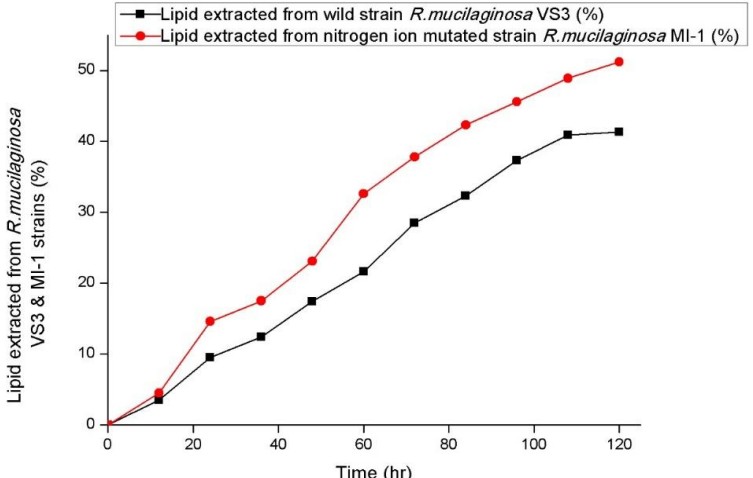

**Figure 6.** A graphical representation of lipid extracted from the VS3 and MI-1 strains of *R. mucilaginosa*.

**Table 4.** Lipid concentration after a number of subcultures.

| S.No | Number of Subculture | Lipid Content (%) from Mutant Strain |
|------|---------------------|--------------------------------------|
| 1 | 6 | 62.3 |
| 2 | 12 | 62.1 |
| 3 | 18 | 61.8 |
| 4 | 24 | 61.8 |
| 5 | 30 | 61.7 |

### 3.5. Fatty Acid Composition Analysis Using GCMS

The composition of fatty acids present in the lipids extracted from *Rhodotorula mucilaginosa* VS3 and MI-1 was analyzed using gas chromatography and mass spectrophotometry. The important fatty acids present in the lipids of the *R. mucilaginosa* VS3 strain were 5.07% Tetradecanoic acid, 9.24% Pentadecanoic acid, 11.24% n-hexadecanoic acid, and 6.85% hexadecanoic acid as represented in Figure 7. In the nitrogen ion beam mutated strain *R. mucilaginosa* MI-1, the properties of the fatty acid and their percentage are recorded as 9.57% Tetradecanoic acid, 11.05% Pentadecanoic acid, 12.36% of n-hexadecanoic acid; 8.17% of hexadecanoic acid; 11.22% of 9-octadecanoic acid and 5.23% 9-octadecanoic acid (Z) Hexyl ester as shown in Figure 8. Therefore, the lipids extracted from *R. mucilaginosa* VS3 and

MI-1 are represented in Table 5 and termed to be a good source for biodiesel production. In contrast to our study, Yuanyuan Cao et al. (2010) have shown more fatty acid compounds such as mystic acid, palmitoleic acid, oleic acid, steric acid, lignoceric acid, and one similar compound, pentadecanoic acid in the *Geotrichum robustum* wild and ion beam mutated strain [29]. Whereas in *R. glutins,* some fatty acid compounds were found to be similar as in the study by Li Shichang et al. (2013), such as pentadecanoic acid, and hexadecenoic acid was found to be similar as our study in both of the VS3 and MI-1 strains [30].

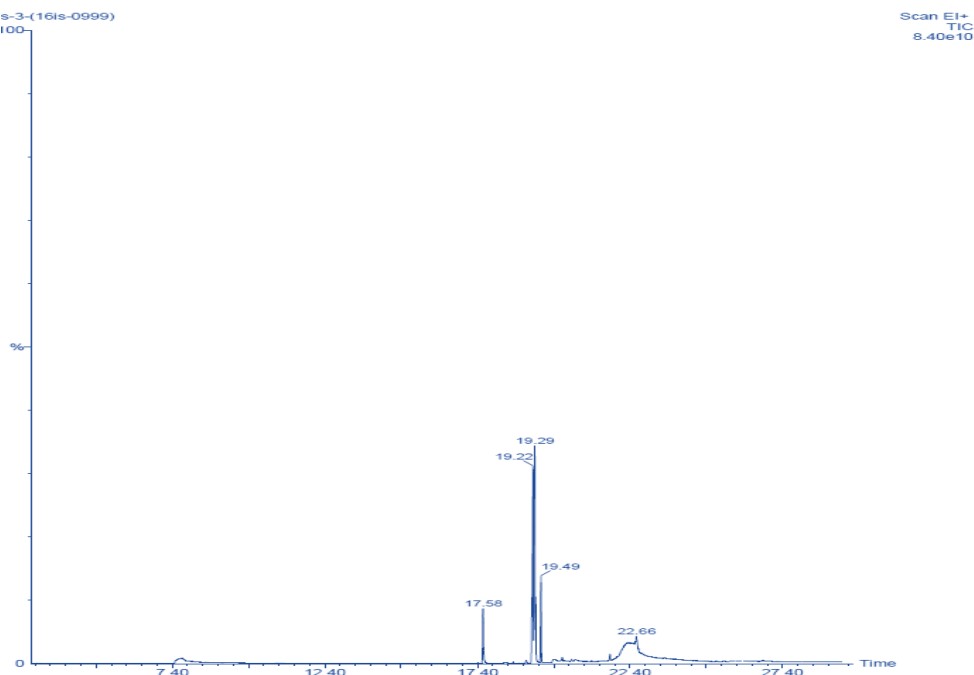

**Figure 7.** A gas chromatographic analysis of lipids produced from the nitrogen ion mutated strain of *R. mucilaginosa* MI-1.

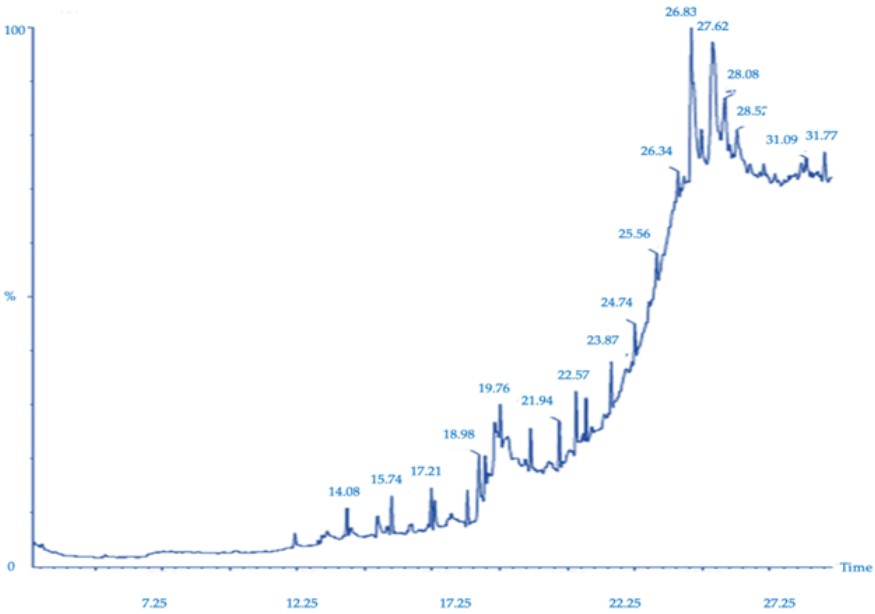

**Figure 8.** Lipid analysis of strain *R. mucilaginosa* VS3 using gas chromatography.

**Table 5.** List of fatty acids present in lipids extracted from strains of *R. mucilaginosa* VS3 and MI-1 analyzed in GCMS.

| Peak No. | Compound Name | RT (Min) | | Fatty Acid (%) | |
|---|---|---|---|---|---|
| | | VS3 | MI-1 | *R. mucilaginosa* VS3 | *R. mucilaginosa* MI-1 |
| 1 | Tetradecanoic acid | 17.58 | 17.21 | 5.07 | 9.57 |
| 2 | Pentadecanoic acid | 19.29 | 18.98 | 9.24 | 11.05 |
| 3 | n-hexadecanoic acid | 19.49 | 19.76 | 11.24 | 12.36 |
| 4 | hexadecanoic acid | 22.66 | 21.94 | 6.85 | 8.17 |
| 5 | 9-octadecanoic acid | - | 25.56 | - | 11.22 |
| 6 | 9-octadecanoic acid (Z) hexyl ester | - | 26.83 | - | 5.23 |

### 3.6. Lipid Analysis Using FTIR

The infrared bonds of lipids produced from strain VS3 attained the wave peak at 1636.45, which represents acid stretching because of the presence of C=O, and the wave attained at 1028.01 indicates acid bending due to the presence of C=O. Therefore, the presence of the carboxylic acid functional group is verified through FTIR analysis and is represented in Figure 9. The extracted lipids from the nitrogen ion irradiated strain MI-1 show infrared regions attaining a peak at 1641.42, indicating acid stretching because of the presence of C=O, and the wavenumber produced at 993.34 signifies acid bending in the presence of C=O. The carboxylic acid functional group present is represented in Figure 10. In accordance with our study, Junying Liu et al. (2013) also stated that signals at 2926 cm$^{-1}$ are better than 1740 cm$^{-1}$ for measuring lipids [40].

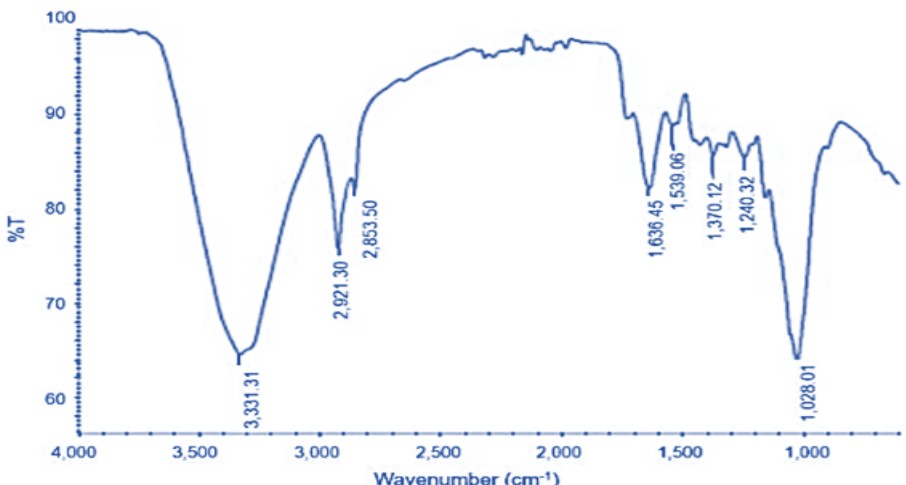

**Figure 9.** FTIR analysis of lipids from *R. mucilaginosa* VS3.

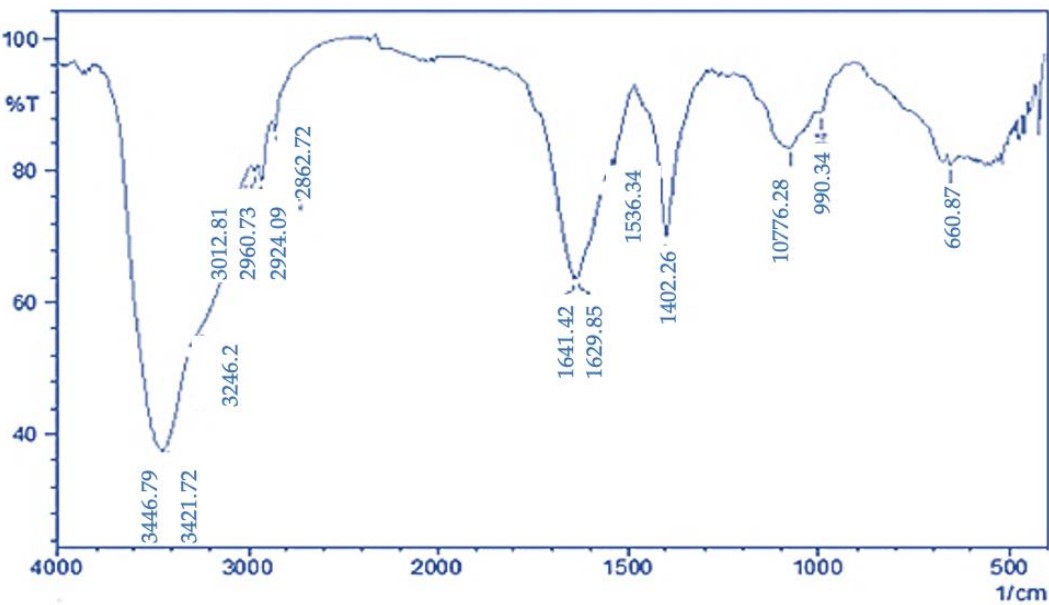

**Figure 10.** FTIR analysis of lipids from *R. mucilaginosa* MI-1.

### 3.7. Biodiesel Properties from the VS3 and MI-1 Strain

The properties of biodiesel produced from the VS3 and MI-1 strains of *R. mucilaginosa* were analyzed and found to be within the ASTM standard limits. The biodiesel density obtained from the wild strain VS3 was 860 Kg/m$^3$, whereas the ion-mutated strain MI-1 showed 867 kg/m$^3$. The specific gravity of biodiesel in the wild strain VS3 was 0.86 and in the nitrogen ion mutated strain was 0.87. The kinematic viscosity of biodiesel in the wild strain VS3 was 5.2 mm$^2$/s and in ion-mutated MI-1 was 5.3 mm$^2$/s as per the ASTM standard. The flash point of the VS3 strain was 160 °C, whereas in the MI-1 ion mutated strain, it was 157 °C. The fire point of biodiesel produced from strain VS3 was 165 °C, and the nitrogen ion irradiated strain MI-1 was 169 °C. The pour point of biodiesel in the wild strain VS3 showed −2 to 8 °C, and the ion mutated strain MI-1 showed −2 to 7 °C. The calorific value of biodiesel produced from the VS3 strain was 37,500 kJ/kg, and the ion-mutated strain MI-1 was 37,450 kJ/kg. The acid value of biodiesel was tested using the free fatty acid titration method from strain VS3, which showed 0.1 mg/KOH, and from the ion mutated strain, which showed 0.2 mg/KOH. The cetane number of biodiesels produced from strain VS3 was 56 min, and in ion mutated strain MI-1 it was 57 min, as represented in Table 6.

**Table 6.** Biodiesel properties from *R. mucilaginosa* VS3 and MI-1 strains.

| S.No | Properties | Units | Indian Standard | American Standard | Biodiesel VS3 | MI-1 | Test Procedure |
|------|-----------|-------|-----------------|-------------------|------|------|----------------|
| 1 | Density | kg/m$^3$ | 860–900 | | 860 | 867 | ASTM D4052-91 |
| 2 | Specific gravity | | | | 0.86 | 0.87 | |
| 3 | Kinematic viscosity | mm$^2$/s | 2.5–6.0 | 1.9–6.0 | 5.2 | 5.3 | ASTM D445 |
| 4 | Flash point | °C | 120 | 150 | 160 | 157 | ASTM D93 |
| 5 | Fire point | °C | 130 | 160 | 165 | 169 | ASTM D93 |
| 6 | Pour point | °C | | | −2 to 8 | −2 to 7 | ASTM D2500 |
| 7 | Calorific value | kJ/kg | 38,500 | - | 37,500 | 37,450 | Demirbas, 2008 |
| 8 | Acid value | Mg/KOH | 0.50 max | 0.80 max | 0.1 | 0.2 | FFA Titration |
| 9 | Cetane number | | 51 min | 47 min | 56 min | 57 min | Krisnangkura, 1986 |

## 4. Conclusions

In this study, it was observed that the wild strain of *R. mucilaginosa* VS3 was subjected to mutation using a low nitrogen ion beam, and the production of lipids can be enhanced by mutating the wild strain of *R. mucilaginosa* VS3 using a low nitrogen ion beam at $1.50 \times 10^{14}$ ions/cm$^2$. We found that the yield of biomass and lipids from the nitrogen ion mutated strain *R. mucilaginosa* MI-1 was high compared to the wild strain *R. mucilaginosa* VS3. Hence, it can be concluded that nitrogen ion beam radiation has a high positive mutation on oleaginous microorganisms. The properties of biodiesel resulted in both the wild and nitrogen ion mutated strains of *R. mucilaginosa* being within the ASTM standards.

**Author Contributions:** J.A.S.T.: investigation; V.S.: conceptualization, methodology, resources. All authors have read and agreed to the published version of the manuscript.

**Funding:** This research received no external funding.

**Data Availability Statement:** No new data were created.

**Conflicts of Interest:** The authors declare no conflict of interest.

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
