# Peer review of "Enhancing the Biodiesel Production by Improving the Yield of Lipids in Wild Strain by Inducing Nitrogen Ion Mutation in Rhodotorula mucilaginosa"

_2036-7481, doi:10.3390/microbiolres14030096_

Round 1
Reviewer 1 Report
The MS entitled "Production of biodiesel by enhancing the yield of lipids in wild strain by inducing nitrogen ion mutation in Rhodotorula mucilaginosa" by Shankar et al. describes production and characterization of oils from wild type and nitrogen ion mutated type strain. It is an interesting read and authors do not oversell their results. Authors will benefit immensely from a proof reader. Entire paragraphs are difficult to grasp and "whereas" and "where" used inappropraitely through out the entire MS. I have attached a few corrections.
While light irradiation of micro-organisms to enhance biodiesel potential is not new, authors fail to highlight the novelty of thier work. Perhaps it is in identifying the adaptability if irradiated strain to a certain environment. Authors need to also improve the background/introduction.
Although results are quite encouraging, work will need significant improvement before publication.

Please proof read entire MS
Author Response
The MS entitled "Production of biodiesel by enhancing the yield of lipids in wild strain by inducing nitrogen ion mutation in Rhodotorula mucilaginosa" by Shankar et al. describes production and characterization of oils from wild type and nitrogen ion mutated type strain. It is an interesting read and authors do not oversell their results. Authors will benefit immensely from a proof reader.
Response: We thank the reviewer for the positive comments on the manuscript.
Entire paragraphs are difficult to grasp and "whereas" and "where" used inappropraitely through out the entire MS.
Response: We corrected the entire manuscript as per the suggestion.
I have attached a few corrections.
Response: We thank the reviewer for the comments about the manuscript. We incorporated all the corrections suggested by the reviewer.
While light irradiation of micro-organisms to enhance biodiesel potential is not new, authors fail to highlight the novelty of thier work. Perhaps it is in identifying the adaptability if irradiated strain to a certain environment. Authors need to also improve the background/introduction. Although results are quite encouraging, work will need significant improvement before publication.
Response: Some modifications in the manuscript have been done which were highlighted in red colour.
Reviewer 2 Report
This manuscript focuses on enhancing the yield of lipids in Rhodotorula mucilaginosa for biodiesel production.
- The manuscript is undoubtedly interesting but poorly written and requires considerable revision.
- The article's title is incorrect; biodiesel is not produced by increasing the yield of lipids. Increasing the yield of lipids can improve the efficiency of biodiesel production. It is desirable to paraphrase more correctly, such as "Boosting Biodiesel Production through Nitrogen Ion Mutation in Rhodotorula mucilaginosa ...".
- There are at least eight grammatical errors in the eight-line abstract!
- There is no mention of statistics / deviations / errors in the results. It is impossible to know whether the difference in results between wild and mutated species is statistically significant.
- There are many inaccurate/incomprehensible wordings of sentences (for example, lines 85-86, 153, 200, 226-227, 246-247, 259, 274, 290, 294, 296, 313, 321-323).
- In many places there are no spaces (for example, lines 101, 139, 142, 165, 177, ….).
- grammatical errors/typos (e.g., litre (line 150), choosen (line 211), …)
- Figures 3, 6 - 9 are not clear, and captions are small or blurry.
- There is constant confusion with designation throughout the m/s (both in the text and in the figures). It should be corrected so that the designation is uniform throughout the text:
· instead "grams, gm, Kg" - "g, kg"
· instead "liter, litre, l, ml" - "L, mL"
· instead "minute, minutes, Min., mins, hours, hrs" -"min, h"
· instead "μ, Cm" - "μm, cm"
· instead " percent, percentage" - "%"
· instead "KJ" - "kJ"
In addition, I marked errors / typos in the text with a marker (see attached file).

Author Response
This manuscript focuses on enhancing the yield of lipids in Rhodotorula mucilaginosa for biodiesel production.
- The manuscript is undoubtedly interesting but poorly written and requires considerable revision.
Response: We thank the reviewer for the positive comments about the manuscript.
- The article's title is incorrect; biodiesel is not produced by increasing the yield of lipids. Increasing the yield of lipids can improve the efficiency of biodiesel production. It is desirable to paraphrase more correctly, such as "Boosting Biodiesel Production through Nitrogen Ion Mutation in Rhodotorula mucilaginosa ...".
Response: We changed the title as per the suggestion.
- There are at least eight grammatical errors in the eight-line abstract!
Response: We apologize for the grammatical errors, We thoroughly checked the manuscript for the grammatical errors.
- There is no mention of statistics / deviations / errors in the results. It is impossible to know whether the difference in results between wild and mutated species is statistically significant.
Response: We included the standard deviations and errors in the results as per the suggestion.
- There are many inaccurate/incomprehensible wordings of sentences (for example, lines 85-86, 153, 200, 226-227, 246-247, 259, 274, 290, 294, 296, 313, 321-323).
Response: We checked the sentences and corrected accordingly.
- In many places there are no spaces (for example, lines 101, 139, 142, 165, 177, ….).
Response: We corrected the mistake and checked for spacing throughout the manuscript.
- grammatical errors/typos (e.g., litre (line 150), choosen (line 211), …)
Response: We corrected the mistakes as per the suggestion.
- Figures 3, 6 - 9 are not clear, and captions are small or blurry.
Response: We changed the figures with better quality.
- There is constant confusion with designation throughout the m/s (both in the text and in the figures). It should be corrected so that the designation is uniform throughout the text:
- instead "grams, gm, Kg" - "g, kg"
- instead "liter, litre, l, ml" - "L, mL"
- instead "minute, minutes, Min., mins, hours, hrs" -"min, h"
- instead "μ, Cm" - "μm, cm"
- instead " percent, percentage" - "%"
- instead "KJ" - "kJ"
Response: As per the suggestion, the manuscript has been corrected with uniform designation.
In addition, I marked errors / typos in the text with a marker (see attached file).
Response: We thank the reviewer for the corrections and suggestions, we corrected the manuscript as per the comments.